# The Potential Role of Cobalt and/or Organic Fertilizers in Improving the Growth, Yield, and Nutritional Composition of *Moringa oleifera*

**Nadia Gad [1], Agnieszka Sekara [2],*** 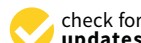 **and Magdi T. Abdelhamid [3],***

[1] Plant Nutrition Department, National Research Centre, 33 El Behouth Street, Cairo 12622, Egypt; dr.nadiagad@yahoo.com
[2] Department of Horticulture, University of Agriculture, 31-120 Krakow, Poland
[3] Botany Department, Ntional Research Centre, 33 El Behouth Street, Cairo 12622, Egypt
* Correspondence: agnieszka.sekara@urk.edu.pl (A.S.); magdi.abdelhamid@yahoo.com (M.T.A.);
  Tel.: +48-12-662-5216 (A.S.); +20-10-0414-5751 (M.T.A.)

**Abstract:** In sustainable farming, the use of organic fertilizers is a costly but environmentally-oriented type of soil–crop system management. Among essential microelements, cobalt (Co) deficiency commonly occurs in arid and semi-arid climatic regions suitable for the growing of moringa (*Moringa oleifera* Lam), an economically important, multipurpose tree. Therefore, in this study, two pot experiments were conducted to identify the interaction effects of Co and organic fertilizers in modifying the growth, yield, and nutritional composition of moringa. Each experiment consisted of 21 treatments as combinations of seven concentrations of Co (0.0, 2.5, 5.0, 7.5, 10.0, 12.5, and 15.0 mg $L^{-1}$) and three organic fertilizers (chicken manure, CM; farmyard manure, FYM; and compost, Comp). Co, at 7.5–12.5 mg $L^{-1}$, in combination with CM, significantly increased the height, leaf number, leaf area, and dry weight of plants, as well as N, P, K, Zn, Cu, protein, total carbohydrate, total soluble solids, total phenolics, carotenoids, and vitamin C in leaves. Co was positively correlated with N, P, K, and the dry weight content in moringa leaves, and this synergistic interaction may underpin the remaining parameters enhanced by Co. The cobalt effect was dose-dependent, so the improved growth, yield, and nutritional composition of moringa can be managed through a proper Co dose in combination with organic fertilizer. Co and organic fertilization could be a promising strategy for improving moringa plant productivity and its biological value in conditions of sandy soils and Co deficiency.

**Keywords:** Co; cobalt; moringa; chemical composition; organic fertilizers; yield

## 1. Introduction

Moringa (*Moringa oleifera* Lam) is the most common crop of the Moringaceae family, comprising 13 species. Moringa is regarded as one of the most valuable and "underutilized" multipurpose tropical crops in the world, as nearly every part of the tree has been used as food or medicine since prehistoric times [1]. Moringa is a perennial tree, often grown by farmers on an annual basis. Its roots, fruits, leaves, and flowers are used as highly nutritive vegetables [2]. The leaves are the plant's most nutritious components and a major source of vitamin A, vitamin $B_2$, vitamin $B_6$, and vitamin C; natural antioxidants like carotenoids, flavonoids, and phenolics; and mineral salts of magnesium, iron, and calcium [3]. Moringa is rich in compounds comprising glucosinolates, which have strong anticancer and antibacterial activity [4]. More research on the agronomy of this crop is needed in order to exploit its potential as food and food ingredients, a source of antioxidant and pharmaceutical products for local and international markets, water purification, and livestock feeds [5].



Moringa is an extremely tolerant species, is little affected by drought, and tolerates a pH of 5.0–9.0 It can grow in regions with an annual precipitation of 250 to 4300 mm and in temperatures from 7 to 48 °C; it even tolerates light frost. Moringa is adapted to a wide range of soil types, but prefers arid, sandy conditions [2,5]. One of the challenges facing world agriculture is increasing the agro biodiversity through the promotion of crops characterized by a high tolerance to infertile soils and vulnerable to eroded soils of drylands [6]. In sandy soils, which are prevalent in arid and semi-arid climatic regions, plant growth is restricted owing to limitations in soil fertility and water availability. Therefore, research work to expand moringa cultivation in deserts and sandy soils is extremely desired in this regard for gaining new knowledge usable for sustainable agriculture and maintaining long-term agricultural output without degrading the natural resources [6].

Recently, due to increases in the price of mineral fertilizers and questions as to their future accessibility, there has been a renewed interest in organic recycling, which can be employed to enhance soil fertility and efficiency. The beneficial effects of organic fertilizers have been reported for several crops, such as faba bean [7], cowpea [8], common bean [9], and wheat [10]. Organic compost waste could therefore be used as a soil nutrient source for crop production in newly reclaimed lands [11–13]. Organic manure has been recorded as an efficient soil modifier that can be used to enhance the nutrient status of the soil and, as a consequence, the development of faba bean [12,14] and *Moringa peregrina* [15]. Sowicki [16] showed that the addition of organic compost considerably improved the dry weight; seed yield; oil content; carbohydrates; and N, P, K, Fe, Mn, Zn, Ni, and cobalt (Co) seed content of the sunflower plant. Managing soil organic matter requires a comprehensive approach to creating a soil environment that can work as a whole to support plant growth and minimize negative environmental impacts [6].

Generally, some trace elements like Co are limited in the diets of much of the world's population, so improving the bioavailability of these elements in food crops is an important strategy for overcoming trace-element deficiencies and improving human health [17]. The cobalt concentration of the earth's crust is around 40 $\mu g\ g^{-1}$, while Co in Egyptian soil ranges from 6.5 to 26.8 mg $kg^{-1}$ of soil, depending on geogenic factors, generally following the relatively higher content of the clay fraction [18]. Co is absorbed by higher plants from the soil in an active way; however, in the xylem, the metal is mainly transported by transpiration flow. Bhattacharya et al. [19] reported that Co bound to Fe and Mn oxides is relatively easily taken up by rice from municipal solid waste compost and cow dung manure. Generally, the enrichment of soil with Co increases Co levels in plants. Bibak [20] reported that providing winter wheat crops with nitrogen improved the absorption of cobalt by plants in sandy loam soil, especially when receiving manure from the farmyard, compared to non-amended plots. Helmy and Gad [21] showed that parsley growth parameters, as well as the essential oil content, increased considerably after cobalt application at 25 mg $kg^{-1}$ soil. Co had an important promotional effect on endogenous hormones in olive crops, as well as parameters of tree growth and yield, and the amount and quality of fruits (i.e., oil content) [22]. However, the mechanisms of Co action leading to increases in crop productivity and quality are not clearly understood.

A better understanding of the beneficial effects of cobalt is important for improving crop productivity and enhancing the nutritional value. The use of cobalt in combination with organic fertilizers can contribute to the conservation of sustainable moringa production by enhancing plant growth, even under stressed environments in sandy soils. The objectives of this research are therefore to determine the viability of *Moringa oleifera* for cultivation in sandy soil and to examine the impact of cobalt on the growth, yield, and nutritional composition of moringa cultivated with various organic fertilizers, i.e., chicken manure, farmyard manure, or compost.

## 2. Materials and Methods

### 2.1. Experimental Procedures

Two pot experiments were conducted at the wire-house of the National Research Centre, Dokki (30°02′17″ N 31°12′40″ E), Giza, Egypt, in two successive 2015 and 2016 seasons to define the best cobalt concentration ranges which give *Moringa oleifera* Lam. with respect to the growth and yield response under three different sources of organic fertilizers.

In the 2015 season (April–August 2015), daytime temperatures ranged from 23.4 to 41.7 °C, with an average of 33.2 ± 3.1 °C, whereas average temperatures at night were 15.5 ± 1.9 °C, with a minimum and maximum of 11.7 and 20.6 °C, respectively. The mean daily relative humidity was 59.4% ± 5.9%, in a range from 42.3% to 69.3%. In the 2016 season (April–August 2016), daytime temperatures ranged from 26.2 to 41.8 °C, with an average of 33.5 ± 2.7 °C, whereas the average temperatures at night were 16.6 ± 2.5 °C, with a minimum and maximum of 11.7 and 23.2 °C, respectively. The mean daily relative humidity was 56.7% ± 9.5% and ranged from 27.0% to 69.4%.

Seeds of *Moringa oleifera* Lam. were kindly provided by The Egyptian Scientific Society of Moringa. The seeds were sown directly in plastic pots (40 cm diameter) on April 16, 2015 and April 18, 2016. Each experiment lasted for 150 days from sowing. Pots of a 10 kg capacity were filled with sandy soil collected from The Experimental Station of National Research Centre, Nubaria, Egypt. The physical and chemical properties of the soil used in the experiment are shown in Table 1.

The treatments comprised two factors. The first factor included three types of organic fertilizers, i.e., chicken manure (CM), farmyard manure (FYM), and compost (Comp). The properties of organic fertilizers are shown in Table 2. The second factor involved seven concentrations of soil-applied cobalt (Co), i.e., 0.0, 2.5, 5.0, 7.5, 10.0, 12.5, and 15.0 mg $L^{-1}$. The experiment consisted of 21 treatments as combinations of seven concentrations of cobalt and three organic fertilizers: (1) 0.0 mg $L^{-1}$ Co + 124 g $pot^{-1}$ CM (T1); (2) 2.5 mg $L^{-1}$ Co + 124 g $pot^{-1}$ CM (T2); (3) 5.0 mg $L^{-1}$ Co + 124 g $pot^{-1}$ CM (T3); (4) 7.5 mg $L^{-1}$ Co + 124 g $pot^{-1}$ CM (T4); (5) 10.0 mg $L^{-1}$ Co + 124 g $pot^{-1}$ CM (T5); (6) 12.5 mg $L^{-1}$ Co + 124 g $pot^{-1}$ CM (T6); (7) 15.0 mg $L^{-1}$ Co + 124 g $pot^{-1}$ CM (T7); (8) 0.0 mg $L^{-1}$ Co + 131 g $pot^{-1}$ FYM (T8); (9) 2.5 mg $L^{-1}$ Co + 131 g $pot^{-1}$ FYM (T9); (10) 5.0 mg $L^{-1}$ Co + 131 g $pot^{-1}$ FYM (T10); (11) 7.5 mg $L^{-1}$ Co + 131 g $pot^{-1}$ FYM (T11); (12) 10.0 mg $L^{-1}$ Co + 131 g $pot^{-1}$ FYM (T12); (13) 12.5 mg $L^{-1}$ Co + 131 g $pot^{-1}$ FYM (T13); (14) 15.0 mg $L^{-1}$ Co + 131 g $pot^{-1}$ FYM (T14); (15) 0.0 mg $L^{-1}$ Co + 264 g $pot^{-1}$ Comp (T15); (16) 2.5 mg $L^{-1}$ Co + 264 g $pot^{-1}$ Comp (T16); (17) 5.0 mg $L^{-1}$ Co + 264 g $pot^{-1}$ Comp (T17); (18) 7.5 mg $L^{-1}$ Co + 264 g $pot^{-1}$ Comp (T18); (19) 10.0 mg $L^{-1}$ Co + 264 g $pot^{-1}$ Comp (T19); (20) 12.5 mg $L^{-1}$ Co + 264 g $pot^{-1}$ Comp (T20); and (21) 15.0 mg $L^{-1}$ Co + 264 g $pot^{-1}$ Comp (T21). Each treatment was replicated four times. The experimental layout was factorial, with two factors in a completely randomized design. The organic fertilizers were mixed thoroughly with the soil of each pot and left for 4 weeks before sowing seeds of moringa. After their emergence, moringa seedlings were thinned, and four plants per pot were left. Seedlings at 21 days after sowing (DAS) were irrigated once with cobalt sulphate salt in the concentrations mentioned.

Irrigation with tap water was practiced to keep soil almost at field capacity by weighing the pot every 2–3 days during the pot experiments. The analytical material was moringa leaves, which were harvested with shoots and cut 10 cm above the soil surface 60, 105, and 150 DAS.

**Table 1.** Physical and chemical characteristics of the soil used in the experiments before cultivation.

| Physical properties | | Value |
|---|---|---|
| Particle size distribution (%) | Coarse sand | 47.78 |
| | Fine sand | 40.08 |
| | Silt | 9.14 |
| | Clay | 3.00 |
| Soil texture | | Fine sand |
| Saturation (%) | | 32.1 |
| Field capacity (%) | | 19.2 |
| Wilting point (%) | | 6.1 |
| Available moisture (%) | | 13.1 |
| Chemical properties | | Value |
| Calcium carbonate (%) | | 4.68 |
| pH (Soil paste) | | 8.9 |
| EC dS m$^{-1}$ (1:2.5) | | 0.44 |
| Soluble cations (meq L$^{-1}$) | $Ca^{2+}$ | 1.97 |
| | $Mg^{2+}$ | 0.86 |
| | $Na^+$ | 1.13 |
| | $K^+$ | 0.42 |
| Soluble anions (meq L$^{-1}$) | $CO_3^{2-}$ | - |
| | $HCO_3^-$ | 0.10 |
| | $Cl^-$ | 2.33 |
| | $SO_4^{2-}$ | 1.95 |
| Available nutrients (mg kg$^{-1}$) | N | 22.2 |
| | P | 8.0 |
| | K | 72.0 |
| | Fe | 4.5 |
| | Mn | 2.7 |
| | Zn | 4.5 |
| | Cu | 5.2 |
| Cobalt (mg kg$^{-1}$) | Soluble Co | 0.35 |
| | Available Co | 4.88 |
| | Total Co | 9.88 |

*2.2. Application of Organic Fertilizers*

Since the soil phosphorus (P) test produced a low value (8.0 mg kg$^{-1}$), application of the three organic fertilizers was based on the nitrogen (N) imposed rate. The N rate was adjusted to 120 kg N ha$^{-1}$ based on the estimated crop available N (kg t$^{-1}$) (Table 2). Consequently, quantities of organic amendments were used with rates of 9.84, 10.43, and 21.05 t ha$^{-1}$, for CM, FYM, and Comp, respectively. The surface area of each pot was 0.1257 m$^2$. Therefore, each pot was amended with 124, 131, and 265 g pot$^{-1}$ for CM, FYM, and Comp, respectively. In CM treatment, each pot was amended with 124 g CM pot$^{-1}$, and contained 2.52 g available N pot$^{-1}$, 2.13 g $P_2O_5$ pot$^{-1}$, and 1.25 g $K_2O$ pot$^{-1}$; in FYM treatment, each pot was amended with 131 g FYM pot$^{-1}$, and contained 2.52 g available N pot$^{-1}$, 1.87 g $P_2O_5$ pot$^{-1}$, and 1.21 g $K_2O$ pot$^{-1}$; and in Comp, each pot was amended with 265 g Comp pot$^{-1}$, and contained 2.52 g available N pot$^{-1}$, 3.31 g $P_2O_5$ pot$^{-1}$, and 2.02 g $K_2O$ pot$^{-1}$.

Organic N must be mineralized before it can be used by the crop. The proportion of organic N in manure that is estimated to be available to the following crop is approx. 25%. Since approx. 12% of the organic N fraction is available in the year following application, the crop available N estimates were adjusted for additional N as follows:

$$\text{Organic N} = \text{Total N} - \text{Ammonium (NH}_4\text{-N)} \tag{1}$$

$$\text{Available Organic N} = \text{Organic N} \times (0.25 + 0.12) \tag{2}$$

$$\text{Retained NH}_4\text{-N} = \text{NH}_4\text{-N} \times \text{Retention Factor} \tag{3}$$

$$\text{Estimated Plant available N} = \text{Available Organic N} + \text{Retained NH}_4\text{-N} \tag{4}$$

$$\text{Application rate (MT) kg t}^{-1} = 120 \text{ kg ha}^{-1}/\text{Plant available N.} \tag{5}$$

**Table 2.** Physical and chemical characteristics of chicken manure (CM), farmyard manure (FYM), and compost (Comp) used in the experiments before moringa cultivation.

| Parameter | CM | FYM | Comp |
|---|---|---|---|
| OM (%) | 36.0 | 32.2 | 24.6 |
| EC (dS m$^{-1}$) | 8.85 | 8.53 | 8.78 |
| pH (1:25) | 6.40 | 6.23 | 8.50 |
| C:N ratio | 7.07 | 6.66 | 8.19 |
| Organic N (kg t$^{-1}$) | 20.3 | 19.3 | 9.5 |
| Total N (kg t$^{-1}$) | 29.6 | 28.1 | 13.9 |
| NH$_4$-N (kg t$^{-1}$) | 9.3 | 8.8 | 4.4 |
| Retained NH$_4$-N (kg t$^{-1}$) | 4.7 | 4.4 | 2.2 |
| Total P (kg t$^{-1}$) | 7.8 | 6.5 | 5.7 |
| Total K (kg t$^{-1}$) | 9.3 | 8.6 | 7.1 |
| Crop available nutrients in first year (kg t$^{-1}$) | | | |
| Available organic N | 7.5 | 7.1 | 3.5 |
| Estimated crop available N | 12.2 | 11.5 | 5.7 |
| Estimated crop available P$_2$O$_5$ | 17.1 | 14.3 | 12.5 |
| Estimated crop available K$_2$O | 10.0 | 9.3 | 7.7 |
| DTPA-extractable (mg kg$^{-1}$): | | | |
| Fe | 5.7 | 515 | 479 |
| Mn | 36.2 | 31.9 | 26.0 |
| Zn | 28.0 | 24.9 | 19.8 |
| Cu | 34.5 | 31.1 | 25.0 |

Since residual manure P is mineralized in the years following application, the next procedures were followed to correct the estimated crop-available P content as follows:

$$\text{Plant available P} = \text{Total P} \times (0.7 + 0.2 + 0.06) \tag{6}$$

$$\text{Plant available P} = \text{Total P} \times (0.7 + 0.2 + 0.06) \tag{7}$$

$$\text{Plant available P}_2\text{O}_5 = \text{Plant available P} \times 2.29. \tag{8}$$

The following procedures were used to correct the estimated crop available K content as follows:

$$\text{Plant available K kg t}^{-1} = \text{Total K} \times 0.9 \tag{9}$$

$$\text{Plant available K}_2\text{O kg t}^{-1} = \text{Crop Available K} \times 1.2. \tag{10}$$

*2.3. Soil Analysis*

The physical and chemical properties of Nubaria soil samples were determined as described by Blackmore et al. [23]. The soil pH; electrical conductivity (EC); cations and anions; organic matter; CaCO$_3$; total nitrogen; and available P, K, Fe, Mn, and Cu contents were measured according to Black et al. [24]. Soluble, available, and total cobalt was determined according to the method described by Cottenie et al. [25].

*2.4. Measurement of Plant Growth Characteristics*

Plants were sampled at 60, 105, and 150 days after sowing and separated into leaves and stems. Plant height (cm), leaf number per plant, and leaf area per plant (cm$^2$) were estimated for each sampling date. Leaf area per plant was determined using a leaf area meter (LI 3100C; LI-COR, Lincoln, NB, USA). Plant fresh weight (FW) (stems + leaves) was recorded. Plant dry weight (DW) (stems + leaves) was recorded after oven drying at 70 °C for 48 h.

*2.5. Analysis of the Nutritional Composition of the Plant*

Fresh samples of leaves were dried at 70 °C to constant moisture before they were ground to a fine powder for analyses of the macro- and micro-nutrient concentration.

For determining N, P, K, Mn, Zn, Cu, Fe, and Co concentrations in moringa leaves, dried ground leaves (1 g) were digested in a mixture of boiling perchloric acid and hydrogen peroxide for 8 h. When the fumes were white and the solution was completely clear, it was cooled to room temperature and filled up to 10 mL with deionized water. Reagent blanks were prepared by carrying out the whole extraction procedure, but in the absence of a sample. Total N was determined using the micro-Kjeldahl method. P was determined colorimetrically using stannous chloride-ammonium molybdate reagent, as described by King [26], after its extraction by sodium bicarbonate, according to Olsen et al. [27]. K was determined using a flame photometer (ELE Flame Photometer, Leighton Buzzard, UK). $Fe^{2+}$, $Mn^{2+}$, $Zn^{2+}$, $Cu^{2+}$, and $Co^{2+}$ concentrations were determined by atomic absorption spectrophotometry, according to Chapman and Pratt [28]. N, P, and K were expressed as g 100 g$^{-1}$ DW, while Mn, Zn, Cu, Fe, and Co were expressed as mg kg$^{-1}$ DW.

The total protein (Pro) concentration was measured with Coomissie Blue, following Bradford [29]. A 100 µL protein sample was mixed at room temperature with 5 mL of water containing 0.12 mM Coomissie Brilliant Blue, 1.02 M ethanol, and 0.87 M orthophosphoric acid. The absorbance at 595 nm was read on a Varian DMS 90 Spectrophotometer (Varian Techtron Pty Ltd., Springvale, Melbourne, Australia) after 5 min and within 1 h after mixing against a reference that omitted the protein.

Total carbohydrates (TC) were extracted from each 0.2 g of dry leaf material placed in a test tube, and 10 mL of 1.0 M sulphuric acid was then added. The tube was sealed and placed in an oven at 100 °C overnight. The solution was then filtered into a 100 mL measuring flask and topped up to the mark with distilled water. The extract was filtered through a Whatman No. 1 filter and the filtrate was oven-dried at 60 °C and then dissolved in a known volume of distilled water. The total carbohydrate concentrations were determined according to Yemm and Willis [30] and expressed in g 100 g$^{-1}$ DW.

The total soluble solid (TSS) concentration in moringa leaves was measured using the procedure described by Dadzie and Orchard [31]. The juice of moringa leaves was prepared by thoroughly mixing 3 g of tissue pulp in 9 mL of distilled water for 2 min and then passing it through filter paper. Following this, one drop of filtrate was placed on the prism of the refractometer and the °Brix value was recorded. The recorded value was multiplied by '3' as the dilution factor.

Total phenolics (TPh) compounds were extracted three times with 100 mL cold 85% (*v/v*) methanol from each 1.0 g DW leaf sample. The methanol extracts were pooled, dried under vacuum, and topped up to a known volume with cold 85% (*v/v*) methanol. A 0.5 mL aliquot of each methanol extract was added to 0.5 mL of Folin–Denis reagent [32], shaken, and allowed to stand for 3 min. One milliliter of saturated sodium carbonate was added to each tube, followed by 3 mL of distilled water, and was then shaken and allowed to stand for 60 min. The absorbance was read at 725 nm to calculate the concentrations of total phenolic compounds (expressed in g 100 g$^{-1}$ DW), according to Daniel and George [33].

Concentrations of carotenoids (Cars) in leaves were determined using a colorimetric method, according to Arnon [34], following an extraction by the homogenization of leaves in 80% acetone. Fresh leaf tissues (0.2 g) were ground in 80% (*v/v*) acetone. After centrifugation at 15,000 g for 10 min, the supernatant was collected and used for absorbance measurements at 470 nm using a

Varian DMS 90 spectrophotometer. Carotenoids' content was calculated with equations suggested by Lichtenthaler [35], and expressed as mg 100 $g^{-1}$ FW.

Vitamin C (VIT-C) was measured according to Mukherjee and Choudhuri [36]. Dried ground leaves (0.5 g) were extracted with 10 mL 6% (*w/v*) trichloroacetic acid. The extract was mixed with 2 mL 2% (*w/v*) dinitrophenylhydrazine, followed by the addition of one drop of 10% (*w/v*) thiourea in 70% (*v/v*) ethanol. The mixture was then boiled for 15 min in a water bath and, after cooling to room temperature, 5 mL 80% (*v/v*) $H_2SO_4$ was added at 0 °C. The absorbance was read at 530 nm. The ascorbate concentration was calculated from a standard curve plotted using known concentrations of ascorbic acid. Vitamin C concentrations were calculated and expressed as mg 100 $g^{-1}$ FW.

### 2.6. Statistical Analysis

A test of normality distribution was carried out according to Shapiro and Wilk's method [37], by using the GenStat 17th Edition software package (VSN International Ltd., Hemel Hempstead, UK). Data were tested for the validation of assumptions underlying the combined analysis of variance by a separate analysis of each season and a combined analysis across the two seasons was then performed if the homogeneity of individual error variances examined by the Levene test [38] was insignificant. The collected data were subjected to a combined analysis of variance (ANOVA) for a factorial experiment with two factors in a completely randomized design [39]. Statistically significant differences between means were compared at $p \leq 0.05$ using Duncan's multiple range test. The statistical analysis was carried out using GenStat 17th Edition (VSN International Ltd., Hemel Hempstead, UK). The correlation coefficient *r* was calculated to determine the relation between the dry weight yield and each of the physiological and chemical traits. Hierarchical cluster analysis was performed on the standardized data using a measure of Euclidean distance and Ward's minimum variance method, as outlined by Ward [40]. Experimental data were also processed for a principal component analysis (PCA) using GenStat 17th Edition (VSN International Ltd., Hemel Hempstead, UK) and the Statistica 12.0 software package (StatSoft Inc., Tulsa, OK, USA), in order to evaluate the existing relationships with original variables.

## 3. Results

The results shown below aimed at studying the possibility of the sustainable cultivation of moringa in greenhouses provided with their required nitrogen, phosphorus, potassium (NPK), and other mineral nutrients from organic sources (chicken manure, farmyard manure, and compost) and improving the moringa plant yield and its nutritional quality through soil enrichment with Co.

### 3.1. Growth and Yield of Moringa Plants

Co treatments at 7.5–12.5 mg $L^{-1}$ resulted in a significant increase of plant height, leaf number, leaf area, and dry weight per plant of moringa for all sampling dates (Table 3). Chicken manure (CM) application resulted in the most significant plant height, leaf number, leaf area, and dry weights, followed by farmyard manure (FYM) and then compost (Comp). However, there was no significant difference between CM and FYM for all mentioned traits. The interaction effects of Co and organic fertilizers showed that Co in concentrations of 7.5, 10.0, and 12.5 mg $L^{-1}$ integrated with CM was the most effective compared to the remaining Co combined with CM, FYM, and Comp for all mentioned traits. The total dry weight (TDW; sum of three sampling dates) increased by 28.2% and 20.1% under CM and FYM, respectively, due to the use of Co at a dose of 10.0 mg $L^{-1}$ compared with the control. However, the increase in the TDW was 35.5% for Comp and Co at a dose of 12.5 mg $L^{-1}$ compared with the control.

**Table 3.** Effects of different rates of cobalt combined with three organic fertilizers—chicken manure (CM), farmyard manure (FYM), and compost (Comp)—on the plant height, leaf number leaf area, and dry weight of plants sampled at 60, 105, and 150 days after sowing (DAS) and the sum of three samplings (total dry weight—TDW) of *Moringa oleifera*.

| Organic Amendment | Cobalt (mg kg$^{-1}$) | Plant Height (cm) | Leaf Number per Plant | Leaf Area per Plant (cm$^2$) | Dry Weight (g per Plant) | | | |
|---|---|---|---|---|---|---|---|---|
| | | | | | 60 DAS | 105 DAS | 150 DAS | TDW |
| CM | 0 | 74.8 ± 0.44 * h–j | 24.8 ± 0.14e–g | 1037 ± 6.1d–f | 19.4 ± 0.11j | 20.2 ± 0.12h–j | 21.3 ± 0.12e–i | 60.9 ± 0.36h–j |
| | 2.5 | 77.9 ± 0.45gh | 27.7 ± 0.16c–e | 1103 ± 6.4c–e | 21.1 ± 0.12f–h | 21.5 ± 0.13e–h | 21.9 ± 0.13d–f | 64.5 ± 0.38e–h |
| | 5 | 80.4f ± 0.47g | 31.7 ± 0.19ab | 1215 ± 7.1b | 21.8 ± 0.13e–g | 22.5 ± 0.13c–f | 22.8 ± 0.13c–e | 67.0 ± 0.39d–f |
| | 7.5 | 94.1 ± 0.55a | 34.7 ± 0.20a | 1334 ± 7.8a | 24.2 ± 0.14a | 26.5 ± 0.16a | 28.0 ± 0.16a | 78.7 ± 0.46a |
| | 10 | 87.5 ± 0.51bc | 33.7 ± 0.20a | 1312 ± 7.7a | 23.1 ± 0.14a–d | 23.7 ± 0.14b–d | 24.0 ± 0.14b–d | 70.7 ± 0.41b–d |
| | 12.5 | 90.5 ± 0.53b | 34.7 ± 0.20a | 1340 ± 7.8a | 23.5 ± 0.14a–c | 24.1 ± 0.14bc | 24.8 ± 0.14bc | 72.3 ± 0.42bc |
| | 15 | 87.6 ± 0.51bc | 31.7 ± 0.19ab | 1215 ± 7.1b | 22.6 ± 0.13b–e | 23.1 ± 0.14b–e | 23.7 ± 0.14b–d | 69.3 ± 0.40b–e |
| | *Mean* | 84.7 ± 1.5A | 31.3 ± 0.78A | 1222 ± 24.5A | 22.2 ± 0.34A | 23.1 ± 0.42A | 23.8 ± 0.46A | 69.0 ± 1.20A |
| FYM | 0 | 73.9 ± 0.43i–k | 24.8 ± 0.14e–g | 1037 ± 6.1d–f | 19.4 ± 0.11j | 20.2 ± 0.12h–j | 21.3 ± 0.12e–h | 60.9 ± 0.36h–j |
| | 2.5 | 75.7 ± 0.44hi | 24.8 ± 0.14e–g | 1041 ± 6.1d–f | 19.7 ± 0.12ij | 20.6 ± 0.12g–i | 21.5 ± 0.13e–g | 61.8 ± 0.36g–j |
| | 5 | 78.3 ± 0.46gh | 25.7 ± 0.15d–g | 1059 ± 6.2d–f | 20.8 ± 0.12g–i | 21.3 ± 0.12f–h | 22.0 ± 0.13d–f | 64.1 ± 0.37f–i |
| | 7.5 | 82.4 ± 0.48ef | 28.7 ± 0.17b–d | 1157 ± 6.8bc | 22.3 ± 0.13c–f | 22.7 ± 0.13c–f | 23.2 ± 0.14c–e | 68.1 ± 0.40c–f |
| | 10 | 86.4 ± 0.50cd | 31.7 ± 0.19ab | 1301 ± 7.6a | 23.7 ± 0.14ab | 24.7 ± 0.14b | 25.6 ± 0.15b | 74.0 ± 0.43b |
| | 12.5 | 85.6 ± 0.50c–e | 29.7 ± 0.17bc | 1217 ± 7.1b | 22.1 ± 0.13d–f | 22.7 ± 0.13c–f | 23.1 ± 0.14c–e | 67.8 ± 0.40c–f |
| | 15 | 83.2 ± 0.49d–f | 26.7 ± 0.16c–f | 1110 ± 6.5cd | 21.6 ± 0.13e–g | 22.2 ± 0.13d–g | 22.8 ± 0.13ab | 66.4 ± 0.39d–g |
| | *Mean* | 80.8 ± 1.0AB | 27.4 ± 0.55AB | 1132 ± 20.7AB | 21.4 ± 0.31AB | 22.0 ± 0.32AB | 22.7 ± 0.31C–E | 66.1 ± 0.93AB |
| Comp | 0 | 62.3 ± 0.36n | 24.8 ± 0.14e–f | 726 ± 4.2i | 14.6 ± 0.08m | 15.1 ± 0.09l | 16.0 ± 0.09l | 45.7 ± 0.27n |
| | 2.5 | 64.9 ± 0.38mn | 19.8 ± 0.12i | 796 ± 4.6hi | 15.6 ± 0.09m | 16.1 ± 0.09kl | 16.7 ± 0.10l | 48.5 ± 0.28mn |
| | 5 | 67.7 ± 0.40lm | 20.8 ± 0.12hi | 872 ± 5.1g | 16.8 ± 0.10l | 17.2 ± 0.10k | 17.7 ± 0.10kl | 51.8 ± 0.30lm |
| | 7.5 | 71.3 ± 0.42j–l | 22.8 ± 0.13g–i | 1028 ± 6.0ef | 18.1 ± 0.11k | 18.7 ± 0.11j | 19.2 ± 0.11h–k | 56.0 ± 0.33kl |
| | 10 | 70.9 ± 0.41kl | 23.8 ± 0.14f–h | 1012 ± 5.9f | 19.6 ± 0.11ij | 19.6 ± 0.11ij | 20.1 ± 0.12f–j | 59.3 ± 0.35i–k |
| | 12.5 | 73.0 ± 0.43i–k | 25.7 ± 0.15d–g | 1080 ± 6.3d–f | 20.2 ± 0.12h–j | 20.7 ± 0.12g–i | 21.3 ± 0.12e–i | 62.2 ± 0.36g–j |
| | 15 | 68.7 ± 0.40l | 22.8 ± 0.13g–i | 839 ± 4.9gh | 19.4 ± 0.11j | 19.1 ± 0.11ij | 19.6 ± 0.11g–k | 58.1 ± j0.34k |
| | *Mean* | 68.4 ± 0.8B | 22.9 ± 0.44B | 908 ± 27.7B | 17.8 ± 0.45B | 18.1 ± 0.41B | 18.7 ± 0.40B | 54.5 ± 1.25B |

* Mean values ± SE within the same column for each trait with the same lower-case letter are not significantly different according to Duncan's multiple range test at $p \leq 0.05$. * Mean values ± SE within the same column for each trait with the same upper-case letter are not significantly different according to Duncan's multiple range test at $p \leq 0.05$.

### 3.2. Mineral Elements of Moringa Leaves

Co treatments at 7.5–12.5 mg L$^{-1}$ resulted in a significant increase of N, P, K, Mn, Zn, Cu, Fe, and Co content in moringa leaves (Table 4). It was noticed that N, P, and K increased by about 50.0%, 96.5%, and 70.8% when Co, at a dose of 7.5 and/or 10.0 mg L$^{-1}$, was used compared with the control. CM application caused the highest significant increase of leaf N, P, K, Zn, and Cu content in moringa leaves, followed by FYM, and then Comp. However, Mn, Fe, and Co were found to be in higher amounts in leaves of plants grown in soil with FYM, followed by CM, and then Comp. Moreover, there was no significant difference between CM and FYM in terms of N, P, K, Mn, Zn, Cu, Fe, and Co contents in moringa leaves. Under the interaction effects of both Co and organic fertilizers, Co applied at a dose of 7.5, 10.0, and 12.5 mg L$^{-1}$ integrated with CM resulted in the highest significant content of N, P, K, Zn, and Cu compared to other Co concentrations combined with CM, FYM, and Comp for all mentioned traits.

However, Co applied in concentrations of 7.5, 10.0, and 12.5 mg L$^{-1}$ with FYM caused the highest Mn content compared to other Co concentrations combined with CM, FYM, and Comp. Regarding the Fe concentration under interaction effects, Co at a dose of 0, 2.5, and 5.0 mg L$^{-1}$, applied together with CM, FYM, and Comp, resulted in the highest content of Fe compared to other treatments. Moringa leaf concentrations of Fe were decreased with an increasing Co rate. Moreover, the behavior of Fe was similar for the three organic fertilizers used. Co in moringa leaves increased gradually with increasing concentrations of Co applied with all organic fertilizers.

### 3.3. Nutritional Composition of Moringa Leaves

Co treatments at a dose of 7.5–12.5 mg L$^{-1}$ resulted in the highest levels of protein (Pro), total carbohydrate (TC), total soluble solids (TSS), total phenolics (TP), carotenoids (Cars), and vitamin C (VIT-C) in moringa leaves (Table 5). The application of Co at a dose of 7.5 mg L$^{-1}$ resulted in increases of Pro, TC, TSS, TP, Cars, and VIT-C content in moringa leaves by 49.9%, 13.4%, 7.7%, 24.3%, 23.7%, and 11.1%, respectively, compared with the control treatment without Co. The application of Co at a dose of 10.0 mg L$^{-1}$ resulted in increases of Pro, TC, TSS, TP, Cars, and VIT-C content in moringa leaves by 36.8%, 13.8%, 6.2%, 21.8%, 20.3%, and 10.1%, respectively, compared with the control treatment without Co. CM application affected the values of Pro, TC, TSS, TP, Cars, and VIT-C most significantly, followed by FYM, and then Comp. Moreover, there was no significant difference between the application of CM and FYM in TC, TSS, TP, Cars, and VIT-C content in moringa leaves. The interaction effects of Co and organic fertilizers showed that Co concentrations at doses of 7.5, 10.0, and 12.5 mg L$^{-1}$ integrated with CM or FYM resulted in the highest significant content of TC, TSS, TP, and VIT-C compared to other Co concentrations combined with CM, FYM, and Comp. Co, at doses of 7.5, 10.0, and 12.5 mg L$^{-1}$, applied with CM, caused the highest significant content of Pro and VIT-A compared to other Co concentrations, combined with CM, FYM, and Comp.

**Table 4.** Effects of different rates of cobalt combined with three organic fertilizers—chicken manure (CM), farmyard manure (FYM), and compost (Comp)—on the element content in *Moringa oleifera*.

| Organic Amendment | Cobalt (mg kg$^{-1}$) | N | P (g 100 g$^{-1}$) | K | Mn | Zn | Cu (mg kg$^{-1}$) | Fe | Co |
|---|---|---|---|---|---|---|---|---|---|
| CM | 0 | 2.60 ± 0.02 $^†$ g–i | 0.17 ± 0.00h–j | 0.95 ± 0.01i–k | 106 ± 0.6b–e | 64 ± 0.4g | 11.6 ± 0.07e | 277 ± 1.6a | 1.34 ± 0.01fg |
| | 2.5 | 2.73 ± 0.02f–h | 0.24 ± 0.00fg | 1.21 ± 0.01e–g | 113 ± 0.7b–d | 66 ± 0.4d–g | 12.4 ± 0.07d | 268 ± 1.6a–c | 1.66 ± 0.01e–g |
| | 5 | 3.50 ± 0.02b–d | 0.28 ± 0.00cd | 1.45 ± 0.01bc | 118 ± 0.7bc | 69 ± 0.4b | 13.4 ± 0.08bc | 263 ± 1.5a–d | 1.86 ± 0.01e–g |
| | 7.5 | 3.89 ± 0.02a | 0.32 ± 0.00ab | 1.62 ± 0.01a | 125 ± 0.7b | 71 ± 0.4a | 14.3 ± 0.08a | 248 ± 1.4c–f | 7.14 ± 0.04b |
| | 10 | 3.89 ± 0.02a | 0.33 ± 0.00a | 1.62 ± 0.01a | 125 ± 0.7b | 71 ± 0.4a | 14.3 ± 0.08a | 256 ± 1.5a–f | 4.31 ± 0.03c–e |
| | 12.5 | 3.72 ± 0.02ab | 0.32 ± 0.00a–c | 1.60 ± 0.01a | 121 ± 0.7b | 69 ± 0.4b | 13.9 ± 0.08ab | 240 ± 1.4e–g | 9.47 ± 0.06ab |
| | 15 | 3.56 ± 0.02bc | 0.27 ± 0.00d–f | 1.57 ± 0.01ab | 119 ± 0.7bc | 68 ± 0.4bc | 13.3 ± 0.08bc | 240 ± 1.4e–g | 11.60 ± 0.07 |
| | *Mean* | 3.41 ± 0.11A | 0.28 ± 0.01A | 1.43 ± 0.05A | 118 ± 1.4AB | 68 ± 0.6A | 13.3 ± 0.21A | 256 ± 3.0AB | 5.34 ± 0.86AB |
| FYM | 0 | 2.60 ± 0.02g–i | 0.17 ± 0.00ij | 0.95 ± 0.01i–k | 106 ± 0.6b–e | 64 ± 0.4g | 11.6 ± 0.07e | 277 ± 1.6a | 1.34 ± 0.01fg |
| | 2.5 | 2.65 ± 0.02gh | 0.19 ± 0.00hi | 1.08 ± 0.01g–i | 108 ± 0.6b–e | 65 ± fg | 12.4 ± 0.07d | 275 ± 1.6a | 1.46 ± 0.01fg |
| | 5 | 2.99 ± 0.02ef | 0.21 ± 0.00hi | 1.13 ± 0.01f–h | 111 ± 0.7b–d | 66 ± 0.4d–f | 13.1 ± 0.08c | 267 ± 1.6a–d | 4.14 ± 0.02c–f |
| | 7.5 | 3.24 ± 0.02de | 0.24 ± 0.00ef | 1.26 ± 0.01d–f | 164 ± 1.0a | 67 ± 0.4c–d | 13.7 ± 0.08a–c | 261 ± 1.5a–e | 6.83 ± 0.04bc |
| | 10 | 3.56 ± 0.02bc | 0.29 ± 0.00b–d | 1.39 ± 0.01cd | 166 ± 1.0a | 67 ± 0.4cd | 13.9 ± 0.08ab | 245 ± 1.4d–f | 10.28 ± 0.06a |
| | 12.5 | 3.56 ± 0.02bc | 0.30 ± 0.00a–d | 1.44 ± 0.01c | 168 ± 1.0a | 68 ± 0.4bc | 14.2 ± 0.08a | 255 ± 1.5a–f | 9.19 ± 0.05ab |
| | 15 | 3.42 ± 0.02cd | 0.28 ± 0.00de | 1.33 ± 0.01c–e | 162 ± 1.0a | 65 ± 0.4e–g | 13.3 ± 0.08bc | 252 ± 1.5b–f | 11.42 ± 0.07a |
| | *Mean* | 3.14 ± 0.08AB | 0.24 ± 0.01AB | 1.22 ± 0.04AB | 141 ± 6.2A | 66 ± 0.3AB | 13.3 ± 0.19A | 262 ± 2.5A | 6.38 ± 0.86A |
| Comp | 0 | 2.08 ± 0.01j | 0.13 ± 0.00j | 0.76 ± 0.01l | 85 ± 0.5ef | 51 ± 0.3j | 9.3 ± 0.05h | 221 ± 1.3g | 1.07 ± 0.01g |
| | 2.5 | 2.37 ± 0.01ij | 0.16 ± 0.00ij | 0.85 ± 0.01kl | 78 ± 0.5f | 53 ± 0.3ij | 8.9 ± 0.05h | 273 ± 1.6ab | 1.43 ± 0.01fg |
| | 5 | 2.50 ± 0.01hi | 0.17 ± 0.00ij | 0.91 ± 0.01j–l | 82 ± 0.5ef | 54 ± 0.3i | 9.3 ± 0.05h | 263 ± 1.5a–d | 1.66 ± 0.01e–g |
| | 7.5 | 2.73 ± 0.02f–h | 0.19 ± 0.00hi | 0.97 ± 0.01i–k | 88d ± 0.5–f | 57 ± 0.3h | 10.0 ± 0.06g | 257 ± 1.5a–f | 4.11 ± 0.02c–f |
| | 10 | 2.86 ± 0.02fg | 0.24 ± 0.00fg | 1.07 ± 0.01g–i | 95c ± 0.6–f | 57 ± 0.3h | 10.5 ± 0.06fg | 238 ± 1.4fg | 10.16 ± 0.05a |
| | 12.5 | 2.99 ± 0.02ef | 0.24 ± 0.00fg | 1.07 ± 0.01g–i | 95 ± 0.6c–f | 58 ± 0.3h | 11.0 ± 0.06ef | 245 ± 1.4d–f | 6.80 ± 0.04bd |
| | 15 | 2.75 ± 0.02f–h | 0.23 ± 0.00fg | 1.01 ± 0.01h–j | 91 ± 0.5d–f | 54 ± 0.3i | 10.2 ± 0.06g | 234 ± 1.4fg | 11.2 ± 0.076a |
| | *Mean* | 2.61 ± 0.07B | 0.19 ± 0.01B | 0.95 ± 0.02B | 88 ± 1.3B | 55 ± 0.5B | 9.8 ± 0.15B | 247 ± 3.7B | 5.2 ± 0.881B |

$^†$ Mean values ± SE within the same column for each trait with the same lower-case letter are not significantly different according to Duncan's multiple range test at $p \leq 0.05$. * Mean values ± SE within the same column for each trait with the same upper-case letter are not significantly different according to Duncan's multiple range test at $p \leq 0.05$.

**Table 5.** Effects of different rates of cobalt combined with three organic fertilizers—chicken manure (CM), farmyard manure (FYM), and compost (Comp)—on bioactive compounds of *Moringa oleifera*.

| Organic Fertilizer | Cobalt (mg kg$^{-1}$) | Proteins | Total Carbohydrates (g 100 g$^{-1}$ FW) | Total Soluble Solids | Total Phenolics (g 100 g$^{-1}$ FW) | Carotenoids (mg 100 g$^{-1}$ FW) | Vitamin C |
|---|---|---|---|---|---|---|---|
| CM | 0 | 8.5 ± 0.05 [†] g–i | 25.8 ± 0.15e | 32.2 ± 0.19d | 2.00 ± 0.01ij | 12.0 ± 0.07h | 18.7 ± 0.11d–f |
| | 2.5 | 9.0 ± 0.05f–h | 26.6 ± 0.16de | 32.6 ± 0.19cd | 2.06 ± 0.01g–i | 12.9 ± 0.08fg | 19.1 ± 0.11c–e |
| | 5 | 11.5 ± 0.07bc | 27.1 ± 0.16cd | 33.2 ± 0.19a–d | 2.17 ± 0.01e–g | 13.7 ± 0.08b–e | 19.7 ± 0.12a–d |
| | 7.5 | 12.8 ± 0.07a | 29.3 ± 0.17a | 34.7 ± 0.20a | 2.48 ± 0.01a | 14.8 ± 0.09a | 20.8 ± 0.12a |
| | 10 | 12.8 ± 0.07a | 28.4 ± 0.17ab | 33.9 ± 0.20a–c | 2.42 ± 0.01a–c | 14.2 ± 0.08a–c | 20.3 ± 0.12a–c |
| | 12.5 | 12.3 ± 0.07ab | 27.9 ± 0.16bc | 33.5 ± 0.20a–d | 2.31 ± 0.01cd | 13.9 ± 0.08b–d | 19.6 ± 0.11a–e |
| | 15 | 11.7 ± 0.07bc | 27. 7 ± 0.16bc | 32.3 ± 0.19d | 2.27 ± 0.01d–f | 13.5 ± 0.08d–f | 18.4 ± 0.11e–g |
| | *Mean* | 11.2 ± 0.37A | 27.6 ± 0.24A | 33.2 ± 0.20A | 2.2 ± 0.044A | 13.6 ± 0.19A | 19.5 ± 0.18A |
| FYM | 0 | 8.5 ± 0.05g–i | 25.7 ± 0.15e | 32.2 ± 0.19d | 2.00 ± 0.01ij | 11.9 ± 0.07h | 18.7 ± 0.11d–f |
| | 2.5 | 8.7 ± 0.05gh | 26.4 ± 0.15de | 32.3 ± 0.19d | 2.04 ± 0.01hi | 12.7 ± 0.07g | 18.9 ± 0.11d–f |
| | 5 | 9.8 ± 0.06ef | 26.9 ± 0.16cd | 33.0 ± 0.19b–d | 2.15 ± 0.01e–h | 13.1 ± 0.08e–g | 19.5 ± 0.11b–e |
| | 7.5 | 10.6 ± 0.06de | 27.7 ± 0.16bc | 33.5 ± 0.20a–d | 2.27 ± 0.01de | 13.6 ± 0.08c–e | 19.8 ± 0.12a–d |
| | 10 | 11.7 ± 0.07bc | 29.3 ± 0.17a | 34.2 ± 0.20ab | 2.44 ± 0.01ab | 14.3 ± 0.08ab | 20.6 ± 0.12ab |
| | 12.5 | 11.7 ± 0.07bc | 28.2 ± 0.17b | 33.6 ± 0.20a–d | 2.38 ± 0.01a–d | 13.1 ± 0.08e–g | 18.8 ± 0.11d–f |
| | 15 | 11.3 ± 0.07cd | 27.6 ± 0.16bc | 32.8 ± 0.19b–d | 2.33 ± 0.01b–d | 12.5 ± 0.07gh | 17.8 ± 0.10f–h |
| | *Mean* | 10.3 ± 0.28B | 27.4 ± 0.25A | 33.1 ± 0.16A | 2.23 ± 0.04A | 13.0 ± 0.17A | 19.2 ± 0.19A |
| Comp | 0 | 6.4 ± 0.04j | 19.4 ± 0.11i | 23.4 ± 0.14i | 1.49 ± 0.01l | 8.9 ± 0.05k | 14.0 ± 0.08i |
| | 2.5 | 7.8 ± 0.05i | 20.3 ± 0.12hi | 25.9 ± 0.15h | 1.76 ± 0.01k | 9.1 ± 0.05k | 15.2 ± 0.09i |
| | 5 | 8.2 ± 0.05hi | 21.5 ± 0.13g | 27.7 ± 0.16fg | 1.90 ± 0.01j | 10.0 ± 0.06j | 16.6 ± 0.10h |
| | 7.5 | 9.0 ± 0.05f–h | 23.0 ± 0.13f | 28.4 ± 0.17ef | 1.95 ± 0.01ij | 10.5 ± 0.06ij | 16.8 ± 0.10h |
| | 10 | 9.4 ± 0.05fg | 21.3 ± 0.12g | 28.4 ± 0.17ef | 2.00 ± 0.01ij | 10.8 ± 0.06i | 17.0 ± 0.10h |
| | 12.5 | 9.8 ± 0.06ef | 22.6 ± 0.13f | 29.3 ± 0.17e | 2.03 ± 0.01hi | 11.1 ± 0.06i | 17.4 ± 0.10gh |
| | 15 | 9.0 ± 0.05f–h | 20.7g ± 0.12h | 26.5 ± 0.16gh | 2.00 ± 0.01 ij | 10.5 ± 0.06ij | 16.7 ± 0.10h |
| | *Mean* | 8.5 ± 0.24C | 21.2 ± 0.26B | 27.1 ± 0.42B | 1.88 ± 0.04B | 10.1 ± 0.17B | 16.3 ± 0.25B |

[†] Mean values ± SE within the same column for each trait with the same lower-case letter are not significantly different according to Duncan's multiple range test at $p \leq 0.05$.

### 3.4. Correlation Matrix

Pearson's correlation coefficients (below diagonal) among all studied attributes of moringa plants grown at 120 kg organic N ha$^{-1}$ of three different organic fertilizer sources, in combination with seven cobalt levels, are shown in Table 6. There were strong relationships between the total dry weight yield and all studied traits ($p \leq 0.01$), which were highly positively associated in a linear way. There was also a strong association between the total dry weight yield and Co ($r = 0.501$; $p \leq 0.01$), which were positively associated in a linear way, as well as between the total dry weight yield and Fe ($r = -0.608$; $p \leq 0.05$), which were highly negatively associated in a linear way.

### 3.5. Linear and Quadratic Response of the Total Dry Weight (TDW) to Cobalt Level

Linear and quadratic responses of the moringa total dry weight per plant to cobalt rates and the three different organic fertilizers are shown in Figure 1. Under CM application, it was expected that the Co increase of 1 mg L$^{-1}$ would cause the TDW to increase by 0.64 g per plant. The R$^2$ value showed that 71% of the variation in TDW could be explained by the quadratic regression model. As seen in Figure 1, for the quadratic curve, TDW = 73.4 was the maximum when $X$ = 9.5, so if Co is applied at a dose of 9.5 mg L$^{-1}$, the TDW is expected to be about 73.3 g per plant. Under FYM application, R$^2$ showed that 70% of the variation in TDW could be explained by the quadratic regression model. Hence, if Co is applied at a dose of 9.9 mg L$^{-1}$, the TDW is expected to be about 69.1 g per plant. Under Comp application conditions, the R$^2$ value showed that 93% of the variation in the total dry weight could be explained by the quadratic regression model. Under Co applied at a dose of 15.0 mg L$^{-1}$, TDW is expected to be about 60.0 g per plant. Considering the average of CM, FYM, and Comp, the $R^2$ value showed that 90% of the variation in the TDW could be explained by the quadratic regression model. With Co applied at the rate of 10.5 mg L$^{-1}$, TDW is expected to be about 58.7 g per plant.

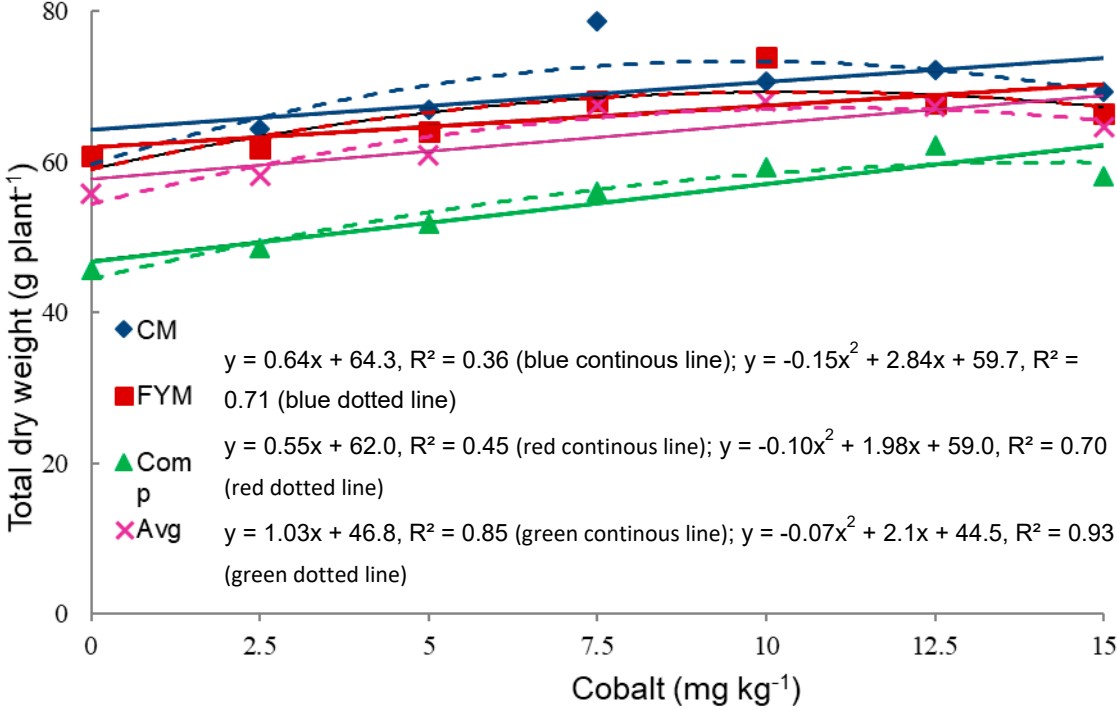

**Figure 1.** Linear and quadratic response of the total dry weight of *Moringa oleifera* grown in combination with three different organic fertilizers, i.e., chicken manure (CM), farmyard manure (FYM), and compost (Comp), and seven cobalt doses, i.e., 0.0, 2.5, 5.0, 7.5, 10.0, 12.5, and 15.0 mg L$^{-1}$. Average of the three organic fertilizers under seven cobalt levels is also shown.

**Table 6.** Pearson's correlation coefficients (below diagonal) among all studied attributes of *Moringa oleifera* grown in combination with three different organic fertilizers—chicken manure (CM), farmyard manure (FYM), and compost (Comp)—and seven cobalt doses, i.e., 0.0, 2.5, 5.0, 7.5, 10.0, 12.5, and 15.0 mg $L^{-1}$.

| | PH | TLNo | TLA | N | P | K | LNo | LA | Cu | Fe | Co | Pro | TC | TSS | TP | Cars | VIT-C |
|---|---|---|---|---|---|---|---|---|---|---|---|---|---|---|---|---|---|
| PH | 1 | | | | | | | | | | | | | | | | |
| TLNo | 0.915 ** | 1 | | | | | | | | | | | | | | | |
| TLA | 0.955 ** | 0.904 ** | 1 | | | | | | | | | | | | | | |
| N | 0.938 ** | 0.892 ** | 0.924 ** | 1 | | | | | | | | | | | | | |
| P | 0.881 ** | 0.869 ** | 0.878 ** | 0.965 ** | 1 | | | | | | | | | | | | |
| K | 0.952 ** | 0.933 ** | 0.921 ** | 0.971 ** | 0.948 ** | 1 | | | | | | | | | | | |
| Mn | 0.744 ** | 0.627 ** | 0.691 ** | 0.694 ** | 0.657 ** | 0.653 ** | 1 | | | | | | | | | | |
| Zn | 0.928 ** | 0.856 ** | 0.826 ** | 0.826 ** | 0.755 ** | 0.862 ** | 0.719 ** | 1 | | | | | | | | | |
| Cu | 0.948 ** | 0.882 ** | 0.915 ** | 0.878 ** | 0.827 ** | 0.893 ** | 0.822 ** | 0.970 ** | 1 | | | | | | | | |
| Fe | −0.045 ns | −0.241 ns | 0.009 ns | −0.190 ns | −0.280 ns | −0.176 ns | 0.001 ns | 0.223 ns | 0.059 ns | 1 | | | | | | | |
| Co | 0.480 ns | 0.375 ns | 0.404 ns | 0.585 ** | 0.624 ** | 0.517 * | 0.496 * | 0.223 ns | 0.379 ns | −0.608 ** | 1 | | | | | | |
| Pro | 0.938 ** | 0.878 ** | 0.927 ** | 0.999 ** | 0.963 ** | 0.967 ** | 0.691 ** | 0.829 ** | 0.877 ** | −0.160 ns | 0.585 ** | 1 | | | | | |
| TC | 0.928 ** | 0.813 ** | 0.903 ** | 0.803 ** | 0.718 ** | 0.823 ** | 0.802 ** | 0.976 ** | 0.963 ** | 0.234 ns | 0.262 ns | 0.806 ** | 1 | | | | |
| TSS | 0.889 ** | 0.750 ** | 0.900 ** | 0.773 ** | 0.698 ** | 0.779 ** | 0.748 ** | 0.967 ** | 0.935 ** | 0.337 ns | 0.225 ns | 0.784 ** | 0.975 ** | 1 | | | |
| TP | 0.943 ** | 0.793 ** | 0.920 ** | 0.938 ** | 0.896 ** | 0.899 ** | 0.791 ** | 0.870 ** | 0.912 ** | 0.015 ns | 0.568 ** | 0.947 ** | 0.888 ** | 0.885 ** | 1 | | |
| Cars | 0.945 ** | 0.883 ** | 0.938 ** | 0.860 ** | 0.803 ** | 0.884 ** | 0.733 ** | 0.976 ** | 0.971 ** | 0.115 ns | 0.316 ns | 0.862 ** | 0.964 ** | 0.954 ** | 0.908 ** | 1 | |
| VIT-C | 0.857 ** | 0.765 ** | 0.892 ** | 0.765 ** | 0.708 ** | 0.765 ** | 0.656 ** | 0.931 ** | 0.898 ** | 0.285 ns | 0.195 ns | 0.776 ** | 0.922 ** | 0.959 ** | 0.869 ** | 0.961 ** | 1 |
| TDW | 0.965 ** | 0.883 ** | 0.956 ** | 0.916 ** | 0.883 ** | 0.909 ** | 0.731 ** | 0.915 ** | 0.937 ** | −0.031 ns | 0.501 * | 0.920 ** | 0.910 ** | 0.907 ** | 0.955 ** | 0.961 ** | 0.917 ** |

** and *—significant at 0.01 and 0.05 levels, respectively; ns—non-significant. PH—plant height; LNo—total leaf number per plant; LA—leaf area; N—nitrogen; P—phosphorus; K—potassium; Mn—manganese; Zn—zinc; Cu—copper; Fe—iron; Co—cobalt; Pro—protein; TC—total carbohydrates; TSS—total soluble solids; TP—total phenolics; Cars—carotenoids; VIT-C—vitamin C; TDW—total dry weight.

### 3.6. Cluster Analysis

Figure 2 shows that the attributes studied could be categorized into three major clusters. The first included the plant height, leaf number per plant, leaf area, N, P, K, Zn, Mn, Cu, Pro, TC, TSS, TP, Cars, and VIT-C, and TDW attributes that had a close similarity and a significant positive correlation. The second cluster included Co, where it was associated with other traits, i.e., plant height, leaf number, leaf area, Zn, Cu, TC, TSS, Cars, and VIT-C in a linear way, but the associations were not significant. The third cluster was Fe, with no significant association with studied traits, i.e., plant height, leaf number, leaf area, N, P, K, Zn, Mn, Cu, Pro, TC, TSS, TP, Cars, VIT-C, and TDW. Moreover, the correlations between Fe and Co or TDW were negative.

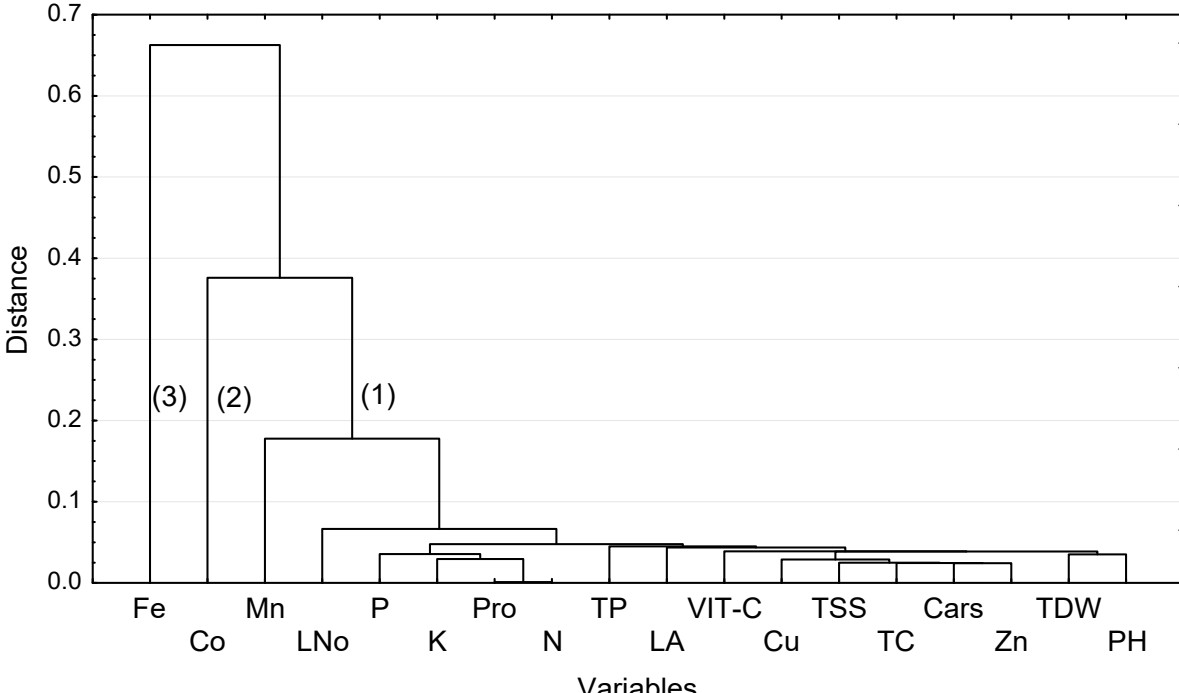

**Figure 2.** Dendrogram showing the extent of distances among different studied attributes. PH—plant height; LNo—total leaf number per plant; LA—leaf area; N—nitrogen; P—phosphorus; K—potassium; Mn—manganese; Zn—zinc; Cu—copper; Fe—iron; Co—cobalt; Pro—protein; TC—total carbohydrates; TSS—total soluble solids; TP—total phenolics; Cars—carotenoids; VIT-C—vitamin C; TDW—total dry weight.

Figure 3 shows that the treatments (T) studied could be categorized into three major clusters, although all treatments showed a close similarity. The first covered T1–T10, T15, and T17–T21 treatments. The second cluster included T11–T14, which were 7.5–15.0 mg $L^{-1}$ Co + 131 g $pot^{-1}$ FYM. The third cluster represented 2.5 mg $L^{-1}$ Co + 264 g $pot^{-1}$ Comp.

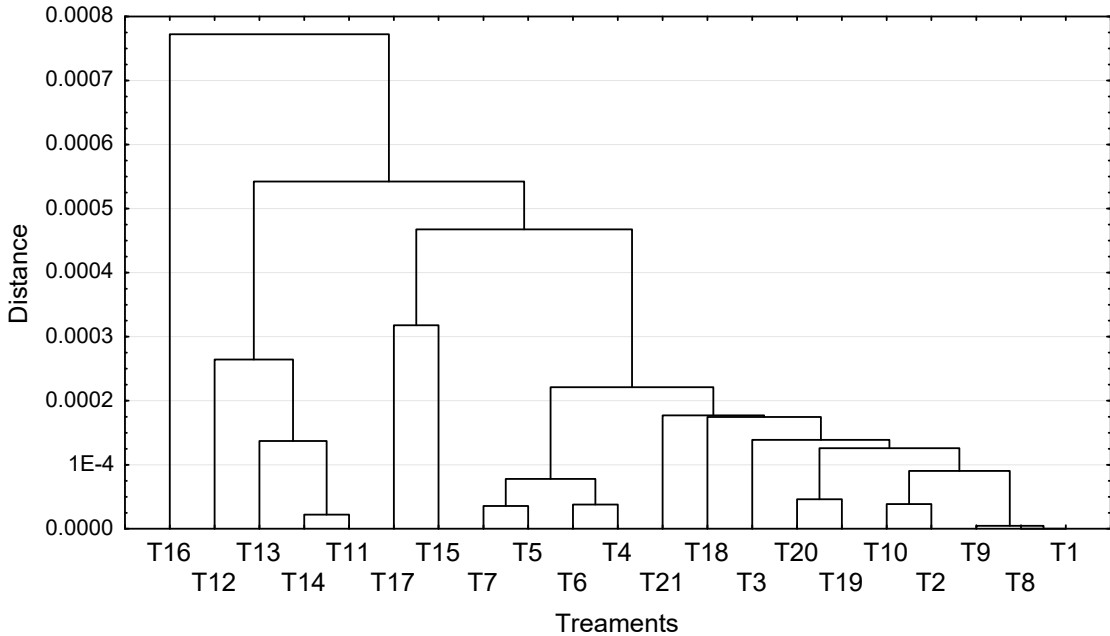

**Figure 3.** Dendrogram showing the extent of distances among different studied treatments. T1 (0.0 mg L$^{-1}$ Co + 124 g pot$^{-1}$ CM); T2 (2.5 mg L$^{-1}$ Co + 124 g pot$^{-1}$ CM); T3 (5.0 mg L$^{-1}$ Co + 124 g pot$^{-1}$ CM); T4 (7.5 mg L$^{-1}$ Co + 124 g pot$^{-1}$ CM); T5 (10.0 mg L$^{-1}$ Co + 124 g pot$^{-1}$ CM); T6 (12.5 mg L$^{-1}$ Co + 124 g pot$^{-1}$ CM); T7 (15.0 mg L$^{-1}$ Co + 124 g pot$^{-1}$ CM); T8 (0.0 mg L$^{-1}$ Co + 131 g pot$^{-1}$ FYM); T9 (2.5 mg L$^{-1}$ Co + 131 g pot$^{-1}$ FYM); T10 (5.0 mg L$^{-1}$ Co + 131 g pot$^{-1}$ FYM); T11 (7.5 mg L$^{-1}$ Co + 131 g pot$^{-1}$ FYM); T12 (10.0 mg L$^{-1}$ Co + 131 g pot$^{-1}$ FYM); T13 (12.5 mg L$^{-1}$ Co + 131 g pot$^{-1}$ FYM); T14 (15.0 mg L$^{-1}$ Co + 131 g pot$^{-1}$ FYM); T15 (0.0 mg L$^{-1}$ Co + 264 g pot$^{-1}$ Comp); T16 (2.5 mg L$^{-1}$ Co + 264 g pot$^{-1}$ Comp); T17 (5.0 mg L$^{-1}$ Co + 264 g pot$^{-1}$ Comp); T18 (7.5 mg L$^{-1}$ Co + 264 g pot$^{-1}$ Comp); T19 (10.0 mg L$^{-1}$ Co + 264 g pot$^{-1}$ Comp); T20 (12.5 mg L$^{-1}$ Co + 264 g pot$^{-1}$ Comp); T21 (15.0 mg L$^{-1}$ Co + 264 g pot$^{-1}$ Comp).

*3.7. Biplot Graph*

The biplot graph (Figure 4) represents the relationship among the moringa treatments using TDW and other studied attributes. The biplot of the mean performance of the moringa data explained 91.32% of the total variation of the standardized data. The first and second principal components (PC1 and PC2) explained 79.48% and 11.84% of variation, respectively. This relatively high proportion reflected the complexity of the relationships among the treatments and the measured traits. The lines perpendicular to the polygon sides facilitate a comparison of neighboring vertex treatments. T4, T5, and T6, which coincided with Co concentrations at rates of 7.5, 10.0, and 12.5 mg L$^{-1}$ integrated with CM, scored the highest significant values, compared to other Co concentrations combined with CM, FYM, and Comp for plant height, leaf number, leaf area, TDW, N, P, K, Zn, Cu, Pro, TC, TSS, TP, Cars, and VIT-C. T11, T12, and T13, which coincided with Co concentrations at rates of 7.5, 10.0, and 12.5 mg L$^{-1}$ integrated with FYM, scored second after CM in terms of the highest values compared to other Co levels combined with CM, FYM, and Comp for most studied traits, as mentioned for CM.

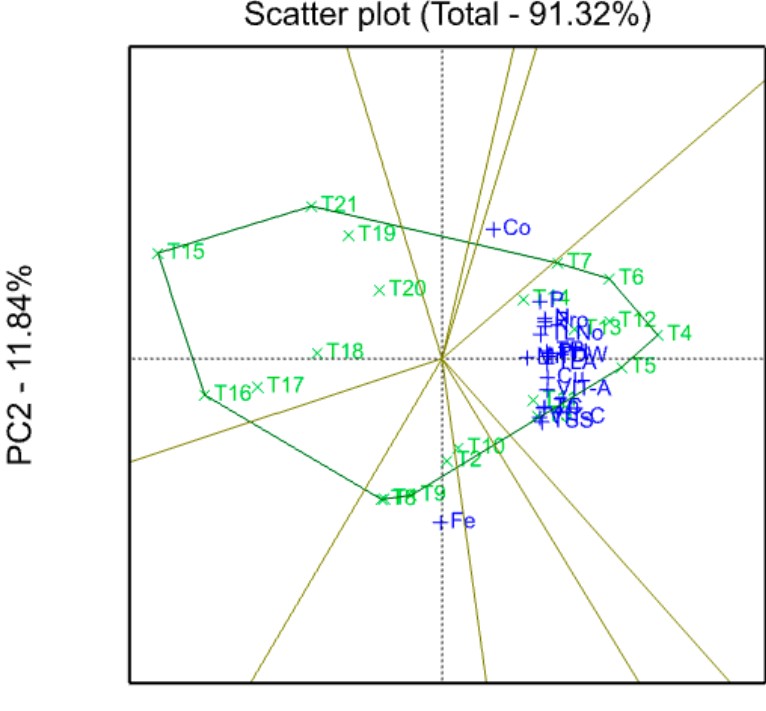

**Figure 4.** Polygon view of the ordination of treatment by trait biplots of principal component analysis outputs. Labels in the graph indicate the investigated treatments (green color) and measured traits (blue color). T1 (0.0 mg $L^{-1}$ Co + 124 g $pot^{-1}$ CM); T2 (2.5 mg $L^{-1}$ Co + 124 g $pot^{-1}$ CM); T3 (5.0 mg $L^{-1}$ Co + 124 g $pot^{-1}$ CM); T4 (7.5 mg $L^{-1}$ Co + 124 g $pot^{-1}$ CM); T5 (10.0 mg $L^{-1}$ Co + 124 g $pot^{-1}$ CM); T6 (12.5 mg $L^{-1}$ Co + 124 g $pot^{-1}$ CM); T7 (15.0 mg $L^{-1}$ Co + 124 g $pot^{-1}$ CM); T8 (0.0 mg $L^{-1}$ Co + 131 g $pot^{-1}$ FYM); T9 (2.5 mg $L^{-1}$ Co + 131 g $pot^{-1}$ FYM); T10 (5.0 mg $L^{-1}$ Co + 131 g $pot^{-1}$ FYM); T11 (7.5 mg $L^{-1}$ Co + 131 g $pot^{-1}$ FYM); T12 (10.0 mg $L^{-1}$ Co + 131 g $pot^{-1}$ FYM); T13 (12.5 mg $L^{-1}$ Co + 131 g $pot^{-1}$ FYM); T14 (15.0 mg $L^{-1}$ Co + 131 g $pot^{-1}$ FYM); T15 (0.0 mg $L^{-1}$ Co + 264 g $pot^{-1}$ Comp); T16 (2.5 mg $L^{-1}$ Co + 264 g $pot^{-1}$ Comp); T17 (5.0 mg $L^{-1}$ Co + 264 g $pot^{-1}$ Comp); T18 (7.5 mg $L^{-1}$ Co + 264 g $pot^{-1}$ Comp); T19 (10.0 mg $L^{-1}$ Co + 264 g $pot^{-1}$ Comp); T20 (12.5 mg $L^{-1}$ Co + 264 g $pot^{-1}$ Comp); T21 (15.0 mg $L^{-1}$ Co + 264 g $pot^{-1}$ Comp). PH—plant height; LNo—leaf number per plant; N—nitrogen; P—phosphorus; K—potassium; Mn—manganese; Zn—zinc; Cu—copper; Fe—iron; Co—cobalt; Pro—protein; TC—total carbohydrates; TSS—total soluble solids; TP—total phenolic; Cars—carotenoids; VIT-C—vitamin C; TDW—total dry weight.

*3.8. Trait Relations (Vector Graph)*

The biplot graph of Figure 5 is a vector drawn from the biplot origin to each marker of the traits to visualize the relationships among them. The results revealed that the investigated traits had strong and positive associations, as shown by the acute angles among their vectors. Meanwhile, there were associations between Co and each of the N, P, K, Mn, protein, and total dry weight (Figure 5). No significant correlation was found between Fe and most of the studied traits. The associations between Fe and Co or the total dry weight were negative.

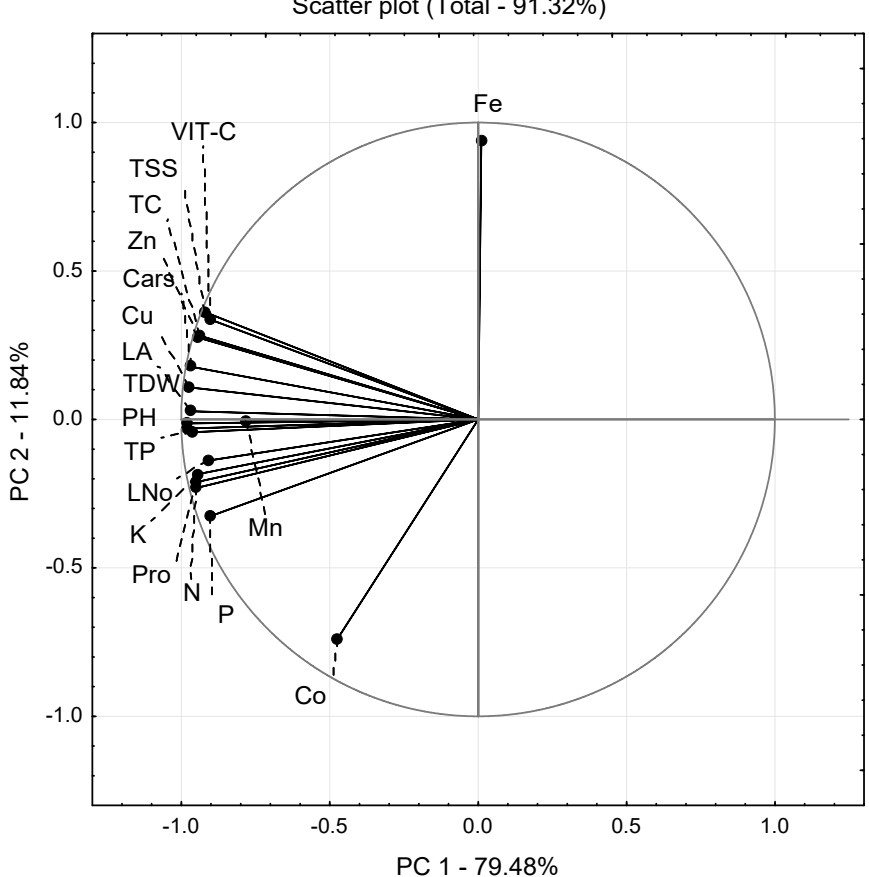

**Figure 5.** Vector view treatment by trait biplot of principal component analysis outputs showing the interrelationship among measured traits for 21 treatments. PH—plant height; LNo—total leaf number per plant; LA—leaf area; N—nitrogen; P—phosphorus; K—potassium; Mn—manganese; Zn—zinc; Cu—copper; Fe—iron; Co—cobalt; Pro—protein; TC—total carbohydrates; TSS—total soluble solids; TP—total phenolics; Cars—carotenoids; VIT-C—vitamin C; TDW—total dry weight.

## 4. Discussion

The higher values of the growth, yield, and yield quality traits obtained in this study with the application of cobalt combined with organic fertilizers were a good indication that moringa plants could respond positively to cobalt combined with organic fertilizers in sandy soils. This positive response covered both morphological and biochemical characteristics, which are important for plant productivity and quality.

Cobalt applied to soil at a dose of 7.5 mg L$^{-1}$ with CM, 10 mg L$^{-1}$ with FYM, and 12.5 mg L$^{-1}$ with Comp resulted in the greatest values of growth parameters of moringa plants. Cobalt at doses higher than those mentioned above showed reduced stimulating effects. The growth-promoting effect of moderate Co doses combined with organic fertilizers confirmed the results of Bibak [20], who stated that winter wheat treated with Co and farmyard manure showed greater growth compared to untreated plants. Gad and Hassan [41] determined that Co and both chicken and farmyard manure had synergistic effects on the fresh and dry weight of tomato shoots and roots compared to control plants, while compost gave the opposite values. Moreover, Gad et al. [22] pointed out that organic fertilizers decreased the soil pH and increased the bioavailability of Co, which improved the olive growth and yield. According to Nanwai et al. [42], the addition of organic manure to sandy soil enhanced the microbial activity and increased the ability to conserve diversification as a final goal.

Generally, using Co at a dose of 7.5–12.5 mg L$^{-1}$ resulted in the greatest values of dry weight per plant and TDW in association with CM. Abd El-Moez and Gad [43] stated that organic cotton

waste compost at 35.7 t ha$^{-1}$ increased cowpea shoot and root fresh and dry weight. Supplementing soil media with 8 mg L$^{-1}$ of Co combined with 23.8 t ha$^{-1}$ of organic cotton waste compost resulted in enhanced cowpea shoot and root growth in the same manner as organic cotton compost alone. Aziz et al. [44] showed that different cobalt treatments significantly increased the growth and yield parameters of lemongrass compared to the control. Gad [45] stated that organic fertilizers significantly increased roselle (*Hibiscus sabdariffa*) growth and yield parameters compared to mineral fertilizers. The application of cobalt at 20 mg L$^{-1}$ with mineral fertilizer in roselle plants caused an increased growth, yield quantity, and content of anthocyanins, flavonoids, and mineral salts. Gad and Hassan [41] found that cobalt at a dose of 7.5 mg L$^{-1}$, together with organic fertilizers, enhanced the growth, yield, nutrients status, and chemical constituents of tomato, as well as tomato's fruit quality. In our study, the increased dry weight yield might be attributed to the increased plant height, leaf number per plant, and total leaf area per plant, which in turn increased the total dry weight. This result was supported by strong evidence obtained for the association between plant height and TDW, leaf number and TDW, and leaf area and TDW, with values of $r = 0.965$, $r = 0.883$, and $r = 0.956$, respectively. Additionally, Gopal et al. [46] showed that the supplementation of tomato growing medium with Co in low doses (<0.5 mM) resulted in a significant increase in chlorophyll content, leading to enhanced productivity.

The application of different rates of cobalt in combination with organic fertilizers improved the N, P, K, Mn, Zn, Cu, and Co content in moringa leaves and CM was the most effective in this respect. The results of the present study confirmed those obtained by Gad and Hassan [41], who stated that chicken manure, followed by farmyard manure, significantly improved the mineral composition in tomato fruits, while agricultural manure was the least effective. The present results indicated that Co application had a significant favorable effect on the mineral status of moringa leaves, with the exception of Fe. This study also convincingly indicated the antagonistic relationships between Co and Fe, which were previously reported by Blaylock et al. [47]. The reason for this was probably the strong association between Co and soil Mn and Fe oxides, making Co unavailable for plants [48], although the present study did not confirm the negative Co–Mn relationship. Moreover, Co that is specifically adsorbed by the oxides could be released as a result of the lowered pH and absorbed by the roots. The present study demonstrated the positive relationship between soil pH and Co in moringa leaves. According to Li et al. [49], the effect of soil pH on the plant Co concentration is masked by the strong fixation of Co by soil Mn and Fe minerals. However, in soils with relatively low contents of Fe and Mn oxides, the pH effect might be more dominant. Increasing the Co level in growing media caused a significantly higher Co concentration in moringa leaves. Gad et al. [22] also found that organic fertilizers increased the availability of indigenous Co, affected by the high pH in Rass Seder conditions. The highest cobalt concentration was recorded in moringa leaves grown in soil amended with FYM (6.38 mg L$^{-1}$), followed by CM (5.34 mg L$^{-1}$) and compost (5.21 mg L$^{-1}$). In light of data published on possible Co toxicity, the present values are within a range safe for the human body. The amount of Co needed for cobalamin synthesis in the human body is very small, and Co dietary uptake should be about 5–40 mg per day [50].

In view of its multiple uses, as well as its culinary and pharmacological values, moringa needs to be widely cultivated in areas where climatic conditions favor its growth [42]. The fertilization treatments investigated in the present study allowed for the identification of the most effective ones for increasing the yield and content of bioactive components in moringa leaves to make it a potential dietary supplement. The application of different rates of cobalt improved the protein, total carbohydrate, total soluble solids, total phenolics, carotenoids, and vitamin C in moringa leaves grown with the addition of organic fertilizers. Gad et al. [51] also reported that Co significantly increased chemical constituents such as the total protein, total soluble solids, total soluble sugars, and vitamin C in dill herb in three harvests during two seasons compared with the control.

The present dendrogram showed that the attributes investigated can be categorized into three major classes. The first cluster included all studied traits except Fe and Co, which were considered the second and third cluster, and the association between Fe and Co or total dry weight were negative.

The biplot of the mean performance of the moringa data explained 91.32% of the total variation of the standardized data. The first and second principal components (PC1 and PC2) explained 79.48% and 11.84%, respectively. This relatively high proportion reflects the complexity of the relationships among the treatments and the measured traits. Yan and Kang [52] mentioned that the first two PC's should reflect more than 60% of the total variation in order to achieve the goodness of fit for the biplot model. The trait's vector length measures the magnitude of its yield effects [53]. Therefore, any two characteristics are positively correlated if the angle between their vectors is an acute angle (<90°), are negatively correlated if their vectors are an obtuse angle (>90°), and have no correlation when their vectors are close to 90° [52]. The presented correlation matrix indicated that the biplot graph was a good substitute procedure for the coefficients of correlation to interpret the interrelations between the studied traits of moringa. It should be noted that the current treatment groups were consistent with those obtained by the cluster analysis. Accordingly, next to or instead of cluster analysis, the biplot graph is considered a successful and effective technique.

## 5. Conclusions

Moringa cultivation can be sustained through application treatments of cobalt combined with organic fertilizers. The results presented here broaden the knowledge on the essential role of Co in plant nutrition, as well as the availability of this metal in sandy soils, depending on organic fertilizers. Chicken manure, in combination with Co, was the most effective concerning the increase in height, leaf number, leaf area, dry weight of moringa plants, as well as N, P, K, Zn, Cu, protein, total carbohydrate, total soluble solids, total phenolic, carotenoids, and vitamin C in leaves. Adequate cobalt and organic fertilization is a promising strategy for improving the productivity, yield, nutritional status, and chemical constituents of moringa. This method of Co biofortification could be positively assessed in terms of safe enrichment of the food chain using this microelement. Combined Co and organic fertilization can be applied even in sandy, easily degradable soils, representing an effective method of sustainable agro ecosystem management for drylands.

**Author Contributions:** This work is a combined efforts of all the authors; Conceptualization and designing the pot experiments, N.G. and M.T.A.; Performing pot experiments and chemical analyses, N.G.; Statistical analysis of data, and producing the presentation tables and figures, A.S. and M.T.A.; Writing the original draft with contributions from all of the authors, A.S. and M.T.A.; Funding acquisition, N.G. and A.S.; Reviewing and editing the whole manuscript, A.S. and M.T.A.

**Funding:** This work was part of Research Project No.10120209, supported by The National Research Centre, Cairo, Egypt. This work was also partially supported by the Ministry of Science and Higher Education of the Republic of Poland.

**Acknowledgments:** Thanks are due to Aboelfetoh Mohamed Abd-Alla, Horticultural crops technology Department, National Research Centre, Cairo, Egypt, for his invaluable comments during conducting pot experiments and writing the manuscript.

**Conflicts of Interest:** The authors declare no conflict of interest.

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
