# Peer review of "The Potential Role of Cobalt and/or Organic Fertilizers in Improving the Growth, Yield, and Nutritional Composition of Moringa oleifera"

_agronomy, doi:10.3390/agronomy9120862_

Round 1

Reviewer 1 Report

The changes that you have made, have improved the quality and understood of article

Reviewer 2 Report

[Introduction] It is edited satisfactorily by articulating the question clearly and providing testable hypotheses. [M & M] It is edited to be a sound M & M. [Results and discussion] This part is pretty good. [Conclusion] The author edited the conclusion part to be compact, concrete and precise.

This manuscript is a resubmission of an earlier submission. The following is a list of the peer review reports and author responses from that submission.

Round 1

Reviewer 1 Report

The manuscript (MS) evaluated the potential role of cobalt and/or organic amendments in improving growth, yield, and nutritional composition of Moringa oleifera. The theme of this manuscript is within the scope of Agronomy. Generally, it is interesting to the readers and conditionally acceptable for publication. But there are many points to be improved and clarified (especially for introduction, materials and methods and conclusion). So, I think some major revisions are needed for considering acceptance of this MS. Please consider the comments below for your revision of the MS.

[Abstract]

There are errors of spelling of words. The author should take English grammar checking with professional natives. In the keywords, the full name (Co) is necessary to describe.

[Introduction]

This study is important for medicinal plants grown by organic-cobalt combination. But the introduction describes a lot of information. The author mentions the importance of moringa plants, the status of sandy soil, the role of Co, and the effects of Co on various plants. But the rationale or question driving the study is not cleared to me. Due to a lot of information, the readers could not catch the points of why organic fertilizers and cobalt applications investigated for Moringa plant. You should omit some lecturers and re-state or organize the information logically or consequently until you get rationale and objective of the study. A better introduction articulates the question clearly and provides testable hypotheses.

The author mentions the use of organic amendments on sandy soil. But you didn’t evaluate the soil fertility changes affected by organic amendments after the experiments. You focus on only plant growth, yield, and nutritional composition. So, I think you should mention just organic “fertilizers”, not “amendment”.

[M & M]

The author uses twenty-one treatments. Among them, 0.0 mg L-1 of Co was applied but the treatment 0.0 g pot-1 of each organic fertilizer was not included. Why? The author applied CM, FYM, and Comp based on the N rate of 120 kg N ha-1. When you applied to the pot experiments, you should describe the N g pot-1 from CM, N g pot-1 from FYM and N g pot-1 from Comp. To clear for the reader, you should add one table in which the name of organic fertilizer, the weight of organic fertilizer (g pot-1), total N, P2O5 and K2O (g pot-1) applied from each organic fertilizer, estimated available N or P2O5 or K2O from each organic fertilizer.

When and how was Co applied in the pot experiment? How to prepare Co solution?

L 147, In Table 1, the units for saturation, field capacity, wilting point, available moisture, cobalt were necessary to add.

L 150 – 166, The author calculated plant-available N, P2O5, K2O from CM, FYM, Comp using the equations. What is the purpose of that calculation? Does the author adjust the N rate based on total N or available N?

L 151, the author describes that the proportion of organic N in manure that is estimated to be available to the following crop is approx. 25%. I don’t agree with that. The amount of available N released from organic fertilizer depends on its total N content or C:N ratio or other. It’s not constant.

L 168 – 189, the author mixed all analysis in paragraph and should separate the paragraph as the measurement of plant growth character, analysis of soil, and analysis of the nutritional composition of the plant.

L 228 – 243, the author analyzed data by combining two years’ data. Why does the author analyze each year's data to know the effects of organic fertilizer on growth, yield and nutritional composition of the plant? The effects of organic fertilizer are different each year, especially in the successive year.

[Results and Discussion]

The results and discussion section are good but there are many mistakes regarding the spelling of words. The author should check English grammar with professional natives. Discuss the main point logically with the direct supporting data. Discuss the facts consequently in a story.

[Conclusions]

The conclusion is too large. The author states uptake of N, P, K, Mn Zn, Cu and Co in moringa plants. Is it uptake or content? All results should not re-state again in the conclusion. It is not necessary. The author should describe the conclusion to be compact, concrete and precise. The conclusion should be drawn based on the main finding and answered the hypothesis. The last conclusion is too general. What kinds of organic fertilizer is effective in combination with Co? With and without Co, how is the plant growth, yield, and nutritional composition?

Author Response

Kraków, 21.11.2019

Dear Editors of the Agronomy

We would like to submit a revised manuscript “The potential role of cobalt and/or organic amendments in improving growth, yield, and nutritional composition of Moringa oleifera” by Nadia Gad, Agnieszka Sekara, Magdi T. Abdelhamid.

We highly appreciate excellent ideas of Reviewers which substantially improved the overall quality of this manuscript. We corrected the text according to the comments of Reviewers. For clarity, we pointed out the main comments of reviewers and our response, and we attached it as a separate file. We highlighted the changes in the text using the "Track Changes" function of Microsoft Word. The text of the revised manuscript is after professional English proofreading by MDPI English Editing Service, the certificate is also attached.

With best regards,

Agnieszka Sekara and co-authors

Reviewer 2 Report

The manuscript have a good novel idea, but a long of then I did not see Co role clearly. All the results and  discusion havent been deeping and I have the feeeling that Co havent strong role to improve the plant growth, and it is due to other factors.

Also, the tables and figures are show without numbres of replicates and standar error, so, it is really confusing and I am not confidance to statiticals result, Sorry!.

I suggest new measured as oxidative status, gas exchanges to improve the results and know Co roles into plants.

Author Response

Dear Editor,

We have attached our response as a separate file.

Sincerely,

Agnieszka Sękara
